

# A new species of *Atrimitra* Dall, 1918 (Gastropoda: Mitridae) from seamounts of the recently created Nazca-Desventuradas Marine Park, Chile

Javier Sellanes[1,2,3], Richard A. Salisbury[4], Jan M. Tapia[5] and Cynthia M. Asorey[3,6]

[1] Departamento de Biología Marina, Facultad de Ciencias del Mar., Universidad Católica del Norte, Coquimbo, Chile
[2] Millennium Nucleus for Ecology and Sustainable Management of Oceanic Islands (ESMOI), Departamento de Biología Marina, Universidad Católica del Norte, Coquimbo, Chile
[3] Sala de Colecciones Biológicas, Universidad Católica del Norte, Coquimbo, Chile
[4] Orma J. Smith Museum of Natural History, The College of Idaho, Caldwell, ID, USA
[5] Programa de Magister en Ciencas del Mar., Universidad Católica del Norte, Coquimbo, Chile
[6] Centro de Estudios Avanzados en Zonas Aridas (CEAZA), Coquimbo, Chile

Corresponding author
Javier Sellanes, sellanes@ucn.cl

## ABSTRACT

We describe *Atrimitra isolata* sp. n. (Gastropoda: Mitridae), collected on the summit of seamounts (~200 m water depth) in the vicinity of Desventuradas Islands, Chile insular territory. Additionally, we provide some insight into the habitat of this new species based on underwater imagery taken with a remotely operated vehicle. *A. isolata* sp. n. is characterized by its small size (up to 26 mm), elongate-ovate shape, solid shell and smooth appearance. It has a base brown color, with some specimens being tan or yellow. It is morphologically related to counterparts from shallow depths on the west coast of North, Central and South America (i.e., *Atrimitra idae*, *Atrimitra orientalis* and *Atrimitra semigranosa*), but has no affinities with species of the family reported from around Easter Island, on the far western side of the Salas y Gómez ridge (e.g., *Strigatella flavocingulata*, *Imbricariopsis punctata* and *Neocancilla takiisaoi*), or with other Indo-Pacific species. The present contribution adds to the knowledge of the poorly studied fauna of the seamounts in the southern portion of the Nazca ridge and easternmost section of the Sala y Gómez ridge, an area characterized by the high degree of endemism of its benthic fauna, and now protected within the large and newly created Nazca-Desventuradas Marine Park.

## INTRODUCTION

In 2015, Chile created the large Nazca-Desventuradas Marine Park (NDMP), covering almost 3,00,000 km$^2$ of this remote part of the SE Pacific. Comprising San Ambrosio and San Félix Islands (known as Desventuradas Islands), and the seamounts located northwest of them, at the intersection of the Salas y Gómez and the Nazca Ridges, this park aims to protect the unique marine fauna inhabiting this area, recognized as a hotspot of

species endemism (*Fernández et al., 2014*; *Friedlander et al., 2016*). As an example, the estimated endemism of fishes, one of the few groups for which enough information exists, is about 40% (*Friedlander et al., 2016*). Conversely, information for invertebrates in the area is sparse. Most of the existing references are associated with research expeditions carried out between 1973 and 1987 by the former Soviet Union, and limited to the area beyond Chilean jurisdiction east of ~83°W (*Mironov & Detinova, 1990*; *Parin, Mironov & Nesis, 1997*). Even with this limited information, endemism estimations in general are outstandingly high, reaching ~46% for the benthic biota (*Parin, Mironov & Nesis, 1997*). For mollusks, these authors report, for the 22 seamounts along the Salas y Gomez and Nazca ridges explored, a total of: one species of Polyplacophora, 27 species of Gastropoda (most of them of the superfamily Conoidea), seven species of Bivalvia, and seven species of Cephalopoda. The latter corresponding to pelagic species, collected most probably during the transit of the trawl nets through the water column. In *Parin, Mironov & Nesis (1997)*, as well as in subsequent malacological studies in the area, no representatives of the family Mitridae have ever been mentioned. However, in the westernmost side of the Salas y Gómez ridge, at Rapa Nui (Easter Island), *Osorio (2018)* mentioned the occurrence of the following three Mitridae species: *Strigatella flavocingulata* (Lamy, 1938), *Imbricariopsis punctata* (Swainson, 1821) and *Neocancilla takiisaoi* (Kuroda, 1959). The two species of the family reported for continental Chile are: *Atrimitra orientalis* (Griffith & Pidgeon, 1834) (see *Marincovich, 1973*) and *Atrimitra semigranosa* (von Martens, 1897) (see *Keen, 1971*), both from northern Chile, ~20–22°S.

In the present study, we revise the Mitridae reported for the region, but with emphasis in continental and insular marine jurisdictional areas of Chile, and describe a new species of *Atrimitra* collected on the summit of seamounts within the NDMP. Insight into the habitat of the new species, based on underwater imagery, is also provided.

## MATERIALS AND METHODS

### Material collection and in situ observations

From October to November 2016, a multidisciplinary oceanographic cruise (CIMAR 22 "Oceanic Islands") was carried out on the research vessel *AGS61Cabo de Hornos*. The aim of the cruise was to study benthic habitats and fauna of unexplored seamounts of the Juan Fernández and Desventuradas Ecoregion (Fig. 1) (*Spalding et al., 2007*; ecoregion number 179). Within the newly created NDMP, six seamounts were visited and six stations were also studied around San Ambrosio and San Felix islands (i.e., Desventuradas Islands) (Fig. 1). Unless weather or sea condition precluded it, the protocol for the benthic survey consisted of a first visual observation of the study site using a ROV (Commander MK2; Mariscope Meerestechnik, Kiel, Germany) equipped with a HD Camcorder (Panasonic SD 909) and laser pointers (10 cm apart), followed by sampling with an Agassiz trawl. The latter consisted of a metal frame with a mouth of 1.5 m × 0.5 m (width × height) fitted with a net of 12 mm mesh at the cod end, operated in 10 min. hauls (bottom contact), at ~3 knots. Collected specimens were preserved in 95% ethanol. Type material as voucher specimens were deposited in the MNHNCL, SCBUCN, ANSP and CIDA, including specimens prepared for scanning electron microscope (SEM)
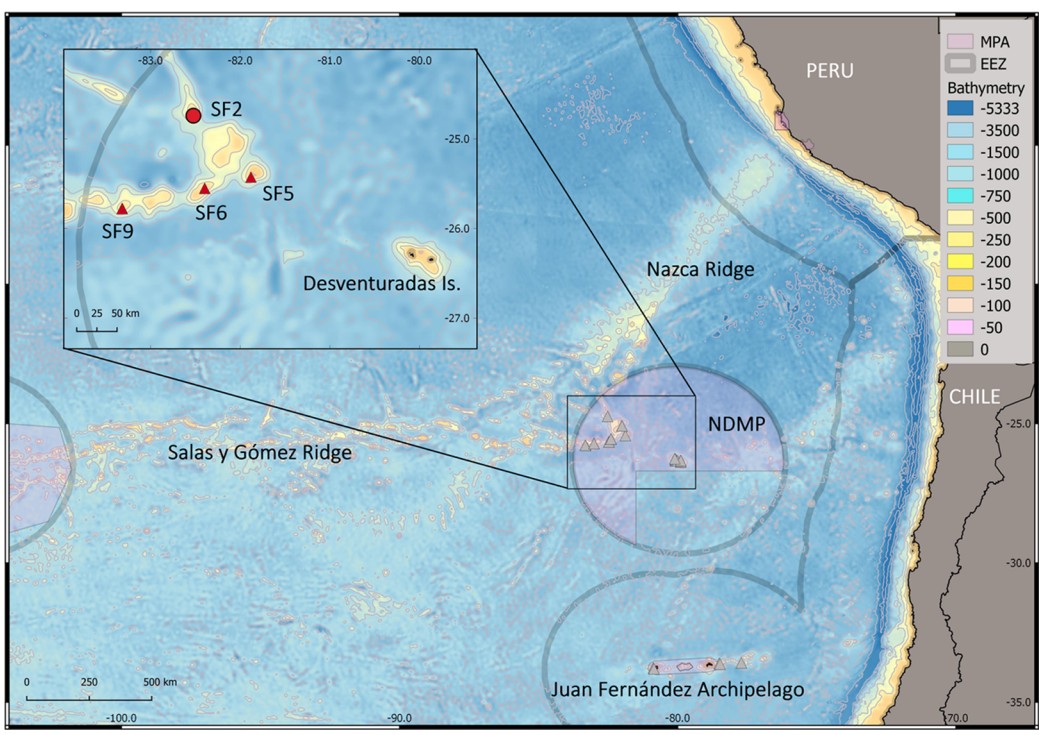

**Figure 1** **Study area.** Map of the study area comprising Desventuradas Islands and seamounts from Salas y Gómez, Nazca Ridge and Juan Fernández Archipelago. Gray triangles: sampled points during CIMAR 22 cruise. Red triangles: seamounts (SF5, SF6 and SF9) where *Atrimitra isolata* sp. n. was collected. Red circle: seamount SF2, in which *Atrimitra isolata* sp. n. was observed in situ. The pink areas represent marine protected areas (MPAs). NDMP, Nazca-Desventuradas Marine Park; EEZ, Exclusive economic zone. Credits for the map: A. Mecho. 

analysis. Sample collection was performed under permission Res. Ext N°41/2016 from SERNAPESCA (Chile) to Universidad Católica del Norte.

The radula and protoconch were examined with a Hitachi SU3500 SEM at the Microscopy Laboratory of the Facultad de Ciencias del Mar, Universidad Católica del Norte, Coquimbo, Chile. A radula from an adult specimen, that was broken for this purpose, was extracted by dissection of the soft parts and cleaned in a 1:50 commercial bleach solution. The examined protoconch was from the same specimen. The radula and the protoconch were dried in a Tousimis, Samdri-780A critical-point dryer using $CO_2$, mounted on bronze stubs and coated with gold in a JEOL JFC-100 evaporator. Description of the radula followed the formula proposed by *Cernohorsky (1970)*, which uses the number of cusps on the lateral and central rachidian plates.

Genomic DNA was extracted from samples SCBUCN 7030, SCBUCN 7031 and SCBUCN 7033 (see type material), from 20 mg of foot tissue of each, and using an E.Z.N.A. ®Tissue DNA kit (Omega, Bio-Tek, Norcross, GA, USA ). In order to amplify partial sequences of the histone 3 (H3) nuclear gene and the mitochondrion cytochrome oxidase I (COI) gene, the pairs of primers H3F (ATGGCTCGTA CCAAGCAGACVGC) and H3R (ATATCCTTRG GCATRATRGTGAC) (*Colgan, Ponder & Eggler, 2000*) and HCO-1490 (GGTCAACAAA TCATAAAGAYATGYG) and

LCO-2198 (TAAACTTCAGGG TGACCAAARAAYCA) (*Folmer et al., 1994*) were used, respectively. The PCR profile for COI started with 5 min at 95 °C, followed by 40 cycles of denaturation at 95 °C (1 min), annealing at 50 °C (1 min), and elongation at 72 °C (2 min), with a final elongation phase at 72 °C (13 min). A similar PCR profile was set for H3 (annealing at 55 °C). Since amplification of the products obtained with both pairs of primers failed, the integrity of genomic DNA samples from all individuals was analyzed by agarose gel electrophoresis, following the procedure described in *Pereira et al. (2011)*. While a tight band (minimal smearing and no banding patterns) of high molecular weight would indicate a high-quality genomic DNA, smearing would indicate degraded DNA, and thus low quality (*Pereira et al., 2011*). In our case, the visualization in the agarose gel showed smearing and no band, suggesting degradation of the DNA, probably caused by sub-optimal preservation of the tissue.

## Nomenclature

The electronic version of this article in Portable Document Format will represent a published work according to the International Commission on Zoological Nomenclature (ICZN) (*International Commission on Zoological Nomenclature, 1999*, *2008*), and hence the new names contained in the electronic version are effectively published under that Code from the electronic edition alone. This published work and the nomenclatural acts it contains have been registered in ZooBank, the online registration system for the ICZN. The ZooBank LSIDs (Life Science Identifiers) can be resolved and the associated information viewed through any standard web browser by appending the LSID to the prefix http://zoobank.org/. The LSID for this publication is: LSID: *Atrimitra isolata* sp. n. urn: lsid: zoobank.org:pub:787A4D2A-260C-49BC-B8B0-0665F2BF6108. The online version of this work is archived and available from the following digital repositories: PeerJ, PubMed Central and CLOCKSS.

## RESULTS

### Systematics account

Superfamily: Mitroidea Swainson, 1831
Family: MITRIDAE Swainson, 1831
Subfamily: Mitrinae Swainson, 1831
Genus: *Atrimitra* Dall, 1918
Type species: *Mitra idae* Melvill, 1893 by original designation.

*Atrimitra isolata* sp. n. Sellanes and Salisbury
Figs. 2A–2H and 3A–3E

**Diagnosis:** Main characteristics of the shell are the small size to 26 mm, elongate-ovate shape, solid, with smooth appearance. Base color brown with some specimens tan or yellow in color.

**Description:** Medium sized shell up to 26 mm, solid, elongate-ovate. Protoconch multispiral, of 4–5 large brown glassy bulbous whorls (Figs. 2D and 3C–3D). Spire whorls convex, post nuclear whorl with numerous weak, beaded, axial ribs, with 3–4 strong, deep

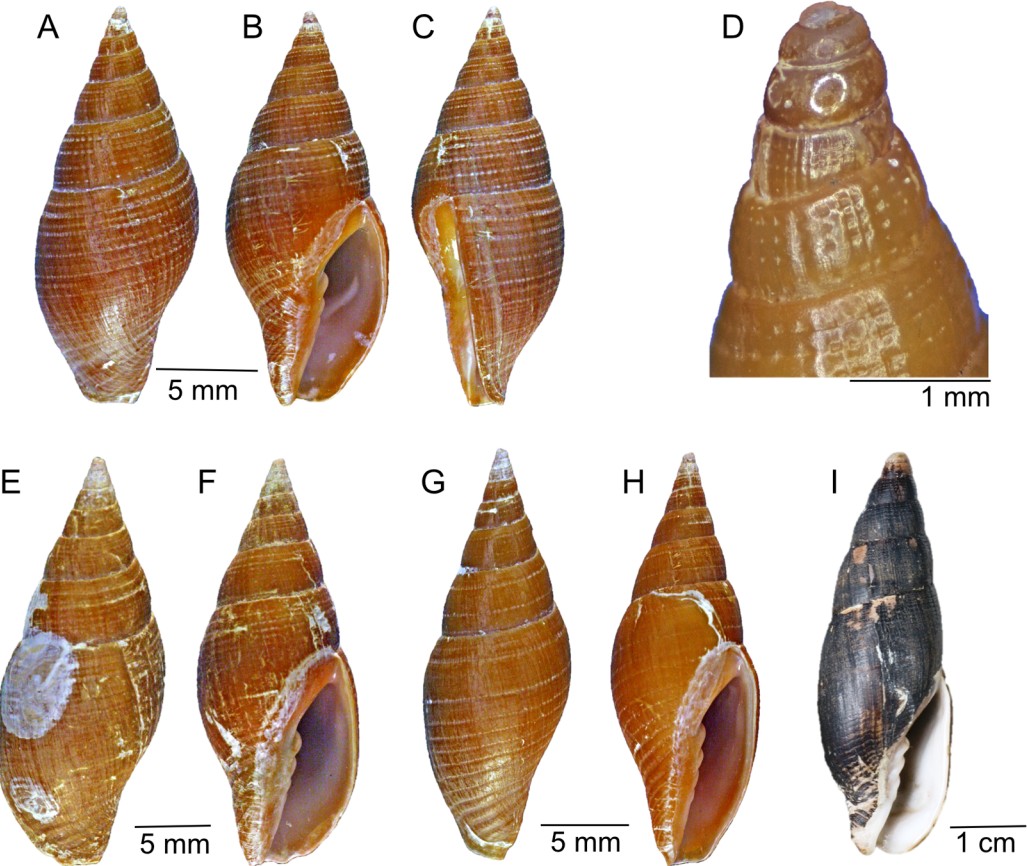

**Figure 2 Type material.** *Atrimitra isolata* sp. n. (A–D) holotype MNHNCL 203730, Seamount SF 9 off Chile, Lat. −25.7774°, Long. −83.163°, 200 m depth. (E and F) Paratype 1 MNHNCL 203731, same as holotype. (G and H) paratype 2 CIDA 126,574, same as holotype. *Atrimitra idae* (I) holotype NMW 1955.158.00100, Point Loma, Baja California, USA. A: abapertural view, B: apertural view, C: side view, D: view of the protoconch and first whorls, E: abapertural view, F: apertural view, G: abapertural view, H: apertural view, I: apertural view.                           

punctate grooves, spiral grooves bisect the axial ribs giving the first whorl a fenestrate sculpture, sculpture changes rapidly on the early whorls, axial ribs become nearly obsolete with spiral punctate grooves varying in number and spacing (Fig. 3E). Penultimate whorl with 6–8 spiral grooves of which 3–4 are deeply punctate, the axial ribs are flattened. Suture distinct but not deeply incised, last adult whorl with 12–14 shallow spiral grooves, half with punctations in the grooves, last adult whorl sculpture changes on the lower half to wide, 10–12 flat spiral cords separated by spiral grooves, the spiral cords are oblique on the fasciole. Aperture of medium width, outer lip gently rounded and smooth, interior of aperture smooth, columella with four columellar folds, siphonal canal short and wide, lacking a siphonal notch. Aperture length greater than half the shell length. Base color brown with some specimens tan or yellow in color. Aperture brown with a faint purple tint. Foot, siphon and eye stalks of the fresh collected animal, white, becoming black when fixed in ethanol. Based on the cusp number the formula of the radula is: 15-5-15, with the lateral rachidian cusp number count +/−1 (Fig. 3E).

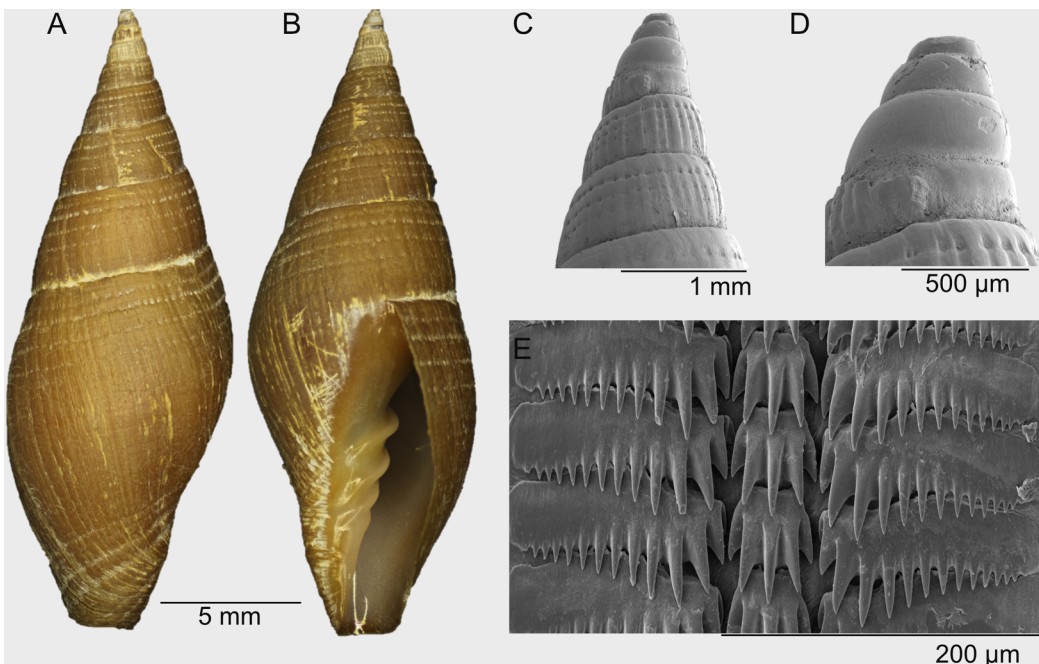

**Figure 3 Radula and protoconch SEMs.** *Atrimitra isolata* sp. n. (A–E) paratype 13 SCBUCN 7030, Seamount SF9 off Chile, Lat. −25.7774°, Long. −83.3163°, 200 m depth. A, abapertural view; B, apertural view; C, SEM side view of the first whorls showing details of the fenestrate sculpture and axial ribs; D, SEM side view of the protoconch; E, SEM of the radula.

**Type material**: Holotype MNHNCL 203730 (Figs. 2A–2D), L: 20.4 mm, W: 7.3 mm, AL: 10.2 mm, seamount off coast of Chile, CIMAR 22 cruise, Station SF9, Lat. −25.7774, Long. −83.163, 27 October, 2016, C22 SSF9 A, trawled, 200 m water depth, lv.

**Additional type material;**

paratype 1 MNHNCL 203731 (Figs. 2E and 2F), L: 25.8 mm, W: 9.2 mm, AL: 13.4 mm, same as holotype, lv.

paratype 2 CIDA 126,574 (Figs. 2G and 2H), L: 21.5 mm, W: 8.1 mm, AL: 11.4 mm, same as holotype, lv.

paratype 3 ANSP 476798, L: 16.1 mm, W: 6.0 mm, AL: 8.1 mm, same as holotype, lv.

paratype 4 MNHNCL 203732, L: 19.1 mm, W: 7.0 mm, AL: 10.8 mm (with predator holes in shell and limpet scars on the columella and aperture), same as holotype, lv.

paratype 5 SCBUCN 7627, L: 11.8 mm, W: 4.9 mm, AL: 6.7 mm, same as holotype, lv.

paratype 6 SCBUCN 6953, L20.4 mm, W: 7.5 mm, same as holotype, d.

paratype 7 SCBUCN 7029, L: 20.1 mm, W: 7.4 mm, same as holotype, lv.

paratype 8 SCBUCN 7033, L: 22.9 mm, W: 8.4 mm, same as holotype (with attached limpet), lv.

paratype 9 SCBUCN 7038, L: 19.6 mm, W: 7.5 mm, Seamount SF5, lv.

paratype 10 SCBUCN 6952a, L: 21.2 mm, W: 7.5 mm, same as holotype, d.

paratype 11 SCBUCN 6952b, L: 21.7 mm, W: 8.0, same as holotype, lv.

paratype 12 SCBUCN 7031, L: 17.1 mm, W: 7.0 mm, Seamount SF6, lv.

paratype 13 SCBUCN 7030 (Figs. 3A–3E), L: 21.4, W: 8.0 mm, same as holotype, lv.
paratype 14 SCBUCN 6946a , L:16.2 mm, W: 6.2 mm, same as holotype, lv.
paratype 15 SCBUCN 6946b, L: 19.1 mm, W: 7.0 mm, same as holotype, lv.
paratype 16 SCBUCN 6946c, L: 20.2 mm, W: 7.6 mm, same as holotype, lv.
paratype 17 SCBUCN 6946d, L: 18.8 mm, W: 7.7 mm, same as holotype (with drill hole), d.
paratype 18 SCBUCN 6947a, L: 22.4 mm, W: 8.8 mm, Seamount SF5, lv.
paratype 19 SCBUCN 6947b, L: 22.9 mm, W: 8.8 mm, Seamount SF5, d.
paratype 20 SCBUCN 6947c, L: 23.4 mm, W: 9.0 mm, Seamount SF5, lv.

**Comparative material:** *Atrimitra idae* (Melvill, 1893), holotype NMW 1955.158.00100, Point Loma, Lower California, USA, *Strigatella coronadoensis* Baker and Spicer, 1930, holotype SDMNH 44409-667, southeastern end of Los Coronados Islands, Lower California, Mexico (Figs. 4A–4C), *Atrimitra semigranosa*, collected near Arica, Parinacota Region, Chile, RAS collection (Figs. 4D–4F), *Atrimitra orientalis*, Lobos de Afuera Islands, Peru, RAS collection (Figs. 4G–4I), two lots of specimens including *Atrimitra orientalis* and *Atrimitra semigranosa*, SCBUCN-7617, Caleta Los Verdes, Iquique, and SCBUCN-7618, El Ñajo, Iquique, Chile.

**Type locality:** Seamount SF9, Lat. −25.7774, Long. −83.3163, Sta. C22SSF9-A, 27 October 2016, at 200 m water depth.

**Distribution and habitat:** Specimen samples come from the summit of three seamounts within the NDMP: SF5 (Lat. −25.4272, Long. −81.8806, 180 m depth), SF6 (Lat. −25.5535, Long. −82.3963, 176 m depth), and SF9 (Lat. −25.7774, Long. −83.3163, 200 m depth). ROV images suggest that the species is also present at nearby seamount SF2 (Lat. −24.7424, Long. −82.5226, 280 m depth). All these seamounts are located within the NDMP.

For the three seamounts on which the species was collected, the summits of two of them (SF6 and SF9) were explored using a ROV. The summit of SF2 was surveyed with the ROV but roughness of the terrain precluded trawling. The bottom at SF6 and SF9 was dominated by coarse sand and the presence of maërl-rhodoliths (Figs. 5A and 5B, respectively), scattered rocky outcrops were also spotted at both sites. Habitat at SF2 differed by the predominance of hard substrates (Fig. 5C). Although about 20 mollusk taxa were found in total at the three collection sites (SF5, SF6 and SF9), species that co-occurred with *A. isolata* sp. n. at all sites were *Architectonica karsteni* Rutsch, 1934 and *Chryseofusus kazdailisi* (Fraussen & Hadorn, 2000).

**Etymology:** From *isolatus* (Latin for isolated) in reference to the remote and isolated geographical location of the four seamounts on which the new species was found.

**Species comparisons:** The holotype of *Atrimitra idae* (Fig. 2I), the type species of the genus *Atrimitra*, measuring 72.1 mm (*Cernohorsky, 1976*) is much larger than the largest recorded specimen of *A. isolata* sp. n. (paratype 1, 25.8 mm). *A. idae* is covered with a thick black periostracum which obscures the sculpture and color pattern of the shell. With the periostracum removed *A. idae*, is brown to tan in color. The early whorls are almost

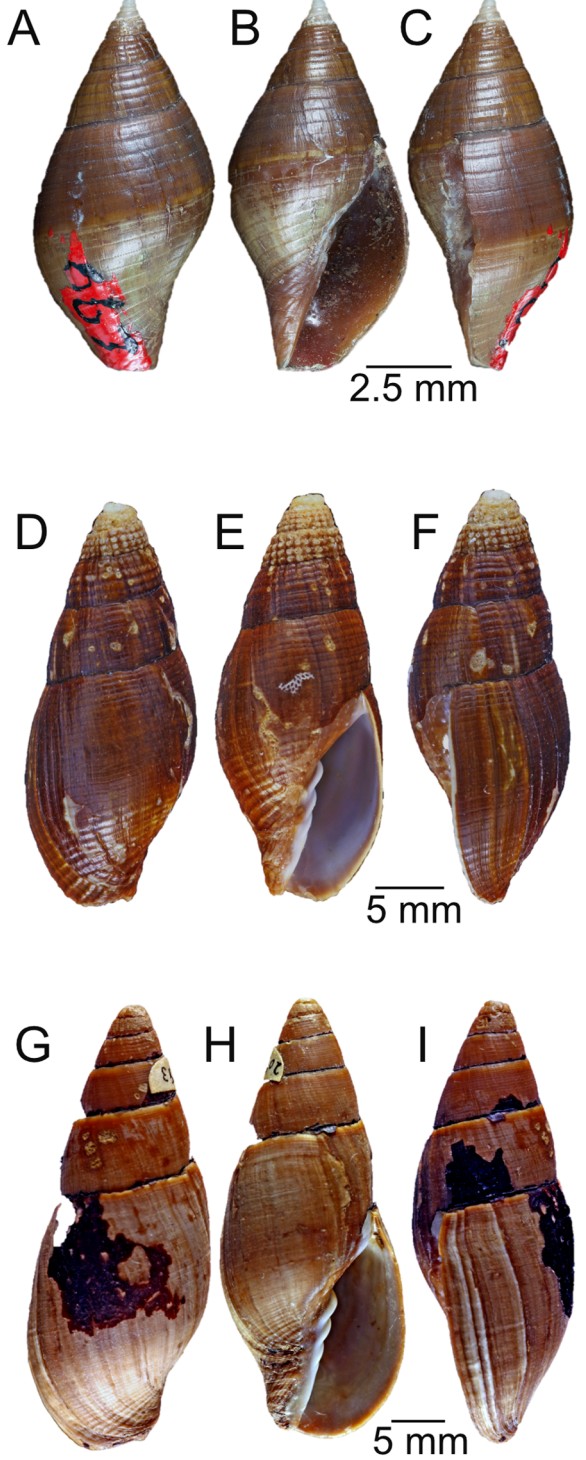

**Figure 4 Comparative species.** Comparative species. (A–C) *Strigatella coronadoensis*, holotype SDMNH 44409-667, southeastern end of Los Coronados Islands, Baja California, Mexico. (D–F) *Atrimitra semigranosa* Arica, Parinacota Region, Chile, RAS collection. (G–I) *Atrimitra orientalis* Lobos de Afuera Islands, Peru, RAS collection. A, abapertural view; B, apertural view; C, side view; D, abapertural view; E, apertural view; F, side view; G, abapertural view; H, apertural view; I, side view.

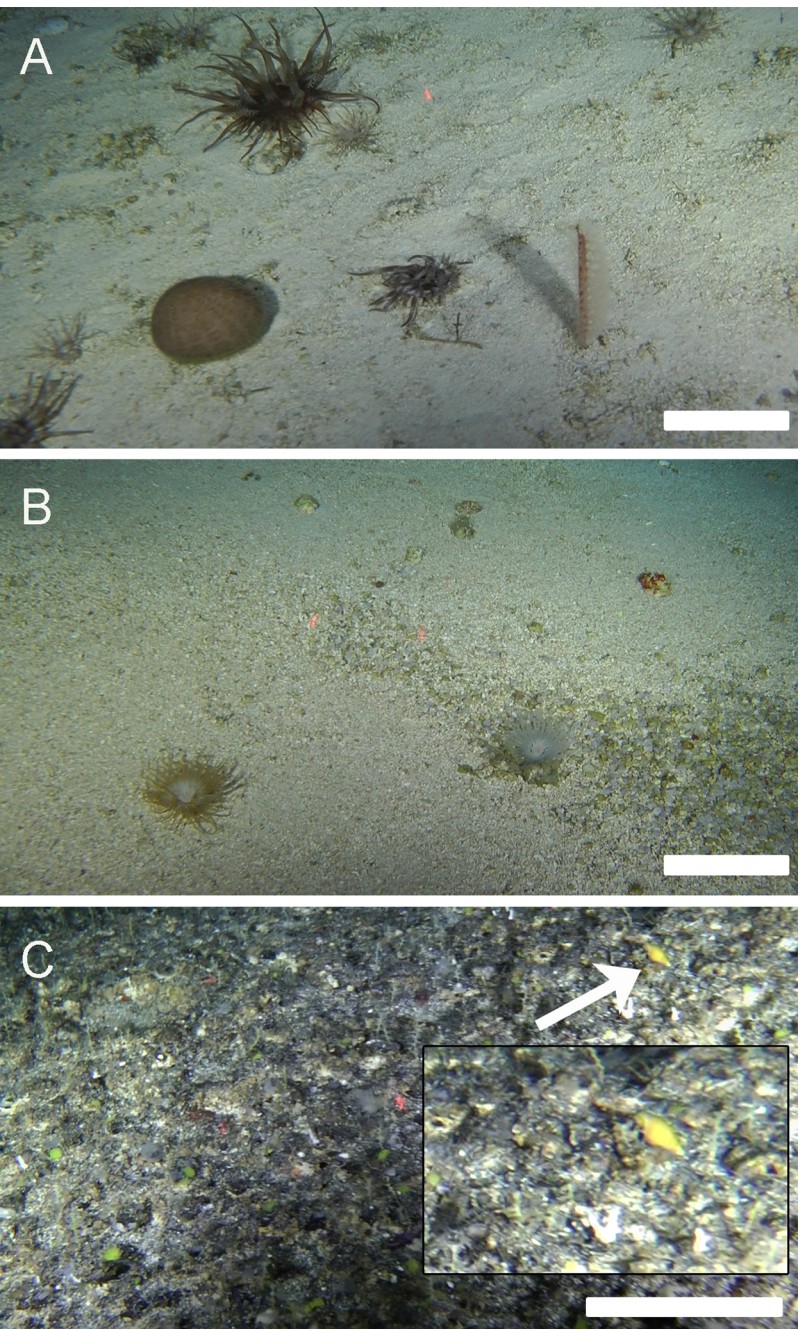

**Figure 5  Habitat.** Images taken with a ROV at the sites where *Atrimitra isolata* sp. n. was spotted within the Nazca-Desventuradas Marine Park. (A) Summit of seamount SF6, 175 m depth, regular continuous homogeneous bottom with little relief, coarse sand dominated by sea pens (*Protoptilum* sp.), sea anemones (*Hormathia* sp. and Cerianthidae) and echinoids (*Stereocidaris nascaensis*). (B) Summit of seamount SF9, 200 m depth, regular continuous homogeneous bottom with little relief, coarse sand and maërl-rhodoliths, dominated by sponges and sea anemones (*Hormathia* sp. and Cerianthidae). (C) Live specimen of *Atrimitra isolata* sp. n. on the summit of seamount SF2, 280 m depth, irregular rock bottom with structures fractured, faulted and folded, dominated by sea pens (*Scleroptilum* sp.) and hydrozoans (*Stylaster marenzelleri*). Image credits: Matthias Gorny, OCEANA.

always eroded and often covered with a thick encrustation. *Cernohorsky (1976)* listed *Strigatella (Atrimitra) coronadoensis* (holotype, Figs. 4A–4C) as a synonym of *Mitra idae*, but this has yet to be confirmed. *Strigatella coronadoensis* has a tiny bullet-shaped, glassy white protoconch of 4–5 whorls. *A. isolata* sp. n. also has a protoconch of 4–5 whorls but these are large, brown, glassy and bulbous. Unlike *A. idae*, the new species has a thin, nearly transparent periostracum, and the sculpture can be seen through it. Sculpture also differs from *A. idae*, which is ornamented with fine spiral grooves, unevenly spaced on the early whorls, with strong axial grooves and growth lines giving the shell a fenestrate appearance. The spiral grooves grow wider on the last adult whorl and the spiral cords also grow wider on the upper part of the last adult whorl. The spiral cords are more uniform in size on the lower part of that adult whorl and not bisected with as many axial grooves or growth lines. *A. isolata* sp. n. is sculptured with widely spaced punctate spiral grooves with fine spiral grooves, usually not punctate that alternate with the deeper punctate grooves. The early whorls are ornamented with shallow axial grooves which form close-set axial ribs. The axial ribs widen and flatten on later whorls. This smoothes the sculpture and makes the shells slippery. The two species live in entirely different habitats, while *A. idae* can be found at depths reachable by scuba and in subtidal habitats such as rocks and rubble, the new species lives at depths between 180 and 280 m on rocky bottoms on seamounts.

Two other Mitridae species have been reported from Chile (*Cernohorsky, 1976*), both formerly in the genus *Mitra* but now placed in *Atrimitra* (*Fedosov et al., 2018*). Both *A. semigranosa* (Figs. 4D–4F) and *A. orientalis* (Figs. 4G–4I), are found in intertidal and subtidal zones associated with rocks, gravel and sand. *Atrimitra semigranosa* can be easily separated from this new species by the pustulate early whorls, and larger size, up to 46 mm. The shell of *A. semigranosa* is covered with a dark brown periostracum, the shell is brown with the early whorls beaded and light brown in color. The beads become obsolete on later whorls with the shell sculptured with spiral cords that are separated by shallow spiral grooves and bisected by axial grooves, giving the mid-whorls a clathrate appearance, the last adult whorl is ornamented with very fine, close-set spiral grooves which grow larger toward the base of the shell. *Atrimitra orientalis* is covered with a thick black periostracum and has a much smoother and larger shell, up to 72 mm, that is gray or light brown in color under the periostracum.

## DISCUSSION

*Atrimitra isolata* sp. n. is one of only a few Mitridae reported from Chilean waters. The new species seems to be isolated from the mainland and so far has been found only on the Nazca Plate, where it lives in deep water associated with seamounts. Since the Nazca and Salas y Gómez ridges are still poorly known in terms of their benthic biodiversity, it is only possible to speculate that the new species might be endemic to the area. The multispiral protoconch of *A. isolata* sp. n. suggests a planktotrophic larval development mode, and thus a high potential for dispersal. On the other hand, physical processes determining connectivity patterns in the area are still poorly known. As an example, it has been suggested that the Humboldt Current System, with characteristic cold

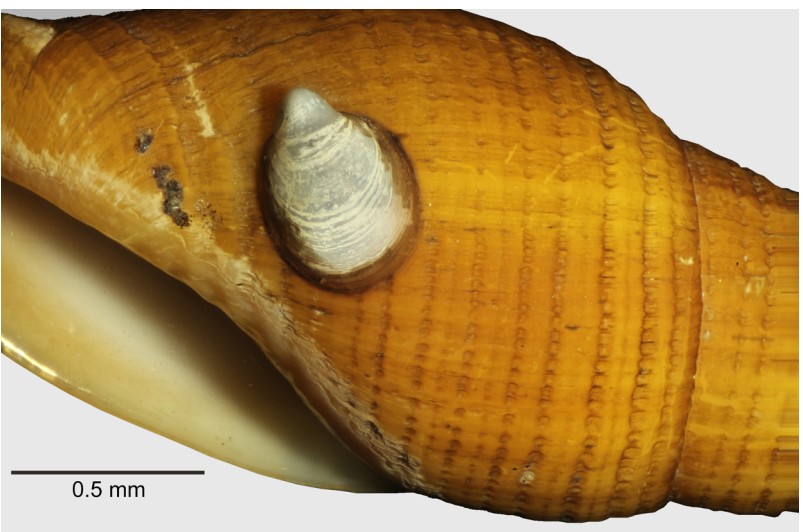

0.5 mm

**Figure 6** **Commensal limpet.** Detail of a hipponicid limpet attached to the shell of *Atrimitra isolata* sp. n., paratype 8 SCBUCN 7033.               

and nutrient-rich waters could be acting as a barrier, at least separating the biota of this area from the South American coast (*Friedlander et al., 2016*). Seamounts are also known to generate particular circulation patterns over their summits, which could be contributing in the retention of locally generated larvae (*Rogers, 2018*). All these factors could be contributing to the isolation of the local fauna and thus to their potential endemism.

The recent publication by *Fedosov et al. (2018)* defining the phylogeny of the Mitridae has indicated that the genus *Atrimitra* Dall, 1918 is represented by several species living along the Western coasts of North, Central and South America. We have chosen to include the new species in *Atrimitra* based on the very fine sculpture of the shell. However, further research, including molecular, analysis is still needed to confidently place the new species within the *Atrimitra* or *Isara* generic units (*Fedosov et al., 2018*). Failure in the extraction of genomic DNA of sufficient quality for sequencing the COI and H3 genes in our specimens could be attributed to deficient tissue preservation. The animal in the preserved specimens was deeply retracted, and considering also that the aperture of the shell is relatively small, probably an amount of ethanol sufficient to avoid DNA degradation did not reach the soft parts.

The number of cusps on the central rachidian plate of the radula is a feature often considered for the taxonomy of Mitridae. For *A. idae*, only drawings of the radula have been published (*Cernohorsky, 1970*, *1976*), and the nonexistence of SEM photos and the little detail presented by the drawings of the radula caused confusion in the cusp formula. Radula of *A. idae* drawings show a formula of 28-6-28 or 28-7-28, with the lateral rachidian plates cusp number +/−3 counts (due to drawing quality). The central rachidian plate in Mitridae often shows two types of formula. The first type presents an even-numbered set of cusps, where each side of the central rachidian plate has the

same number and size of cusps (R.A. Salisbury, 2019, personal observations). The second type presents a longer central cusp with shorter lateral cusps on each side. This type has an odd number of cusps and *A. isolata* sp. n. is an example of this central rachidian type which has five cusps. However, there are not enough SEM images of radulae of this type (see *Fedosov et al., 2018*) to make any decisions as to the importance of the cusp count on the central rachidian plate.

It is interesting to note that species of the family Mitridae found around Easter Island, *Strigatella flavocingulata* (Lamy, 1938), *Imbricariopsis punctata* (Swainson, 1821) and *Neocancilla takiisaoi* (Kuroda, 1959), reviewed in *Osorio (2018)*, on the far Western side of the Salas y Gómez ridge, are all Indo-Pacific species, with ranges across the Indian and Pacific Ocean. The new species has no morphological affinities with them and available evidence suggests that it is found only on seamounts of this region, which hosts a fauna characterized by the high levels of endemism (*Friedlander et al., 2016*).

An interesting ecological observation is that some specimens of *A. isolata* sp. n. show drill holes, perhaps from Muricidae, Naticidae or other predators. Shells of live and dead specimens sometimes present scars from a hipponicid limpet (Fig. 6). Although, we cannot confirm identity, similar limpets are also found attached to spines of the urchin *Stereocidaris nascaensis* (J.M. Tapia, 2019, personal observations), suggesting that the relationship with *A. isolata* sp. n. is just an opportunistic commensalism. Regarding potential food sources of *A. isolata* sp. n., it has been observed that rhodoliths recovered from SF6 and SF9 seamounts were profusely bored by sipunculans of the genus *Aspidosiphon* (J.M. Tapia, 2019, personal observations). Sipunculans have been often reported as a prey for Mitridae (*Ponder, 1998*). For further details of the habitat and ecological aspects of these seamounts, refer to *Easton et al. (2019)*.

## CONCLUSIONS

We describe *A. isolata* sp. n. from seamounts near Desventuradas Islands, at the intersection of the Nazca and Salas y Gómez Ridges. Although the region is still poorly studied in terms of its benthic biodiversity, the new species has so far been found only in this area. Available evidence suggests that the new species is more closely related to eastern Pacific Mitridae and not to other central Pacific or Indic Ocean counterparts. Further molecular analysis is still needed to properly place the new species within the *Atrimitra* or *Isara* generic units. The present contribution adds to the knowledge of the fauna of seamounts of the Salas y Gómez and Nazca Ridges, an area known by its high levels of endemism, and part of which is now protected within the large and newly created NDMP.

## ABBREVIATIONS

| | |
|---|---|
| **AL** | Aperture length (mm) |
| **ANSP** | Academy of Natural Sciences of Drexel University, Philadelphia, USA |
| **CIDA** | Orma J. Smith Museum of Natural History, The College of Idaho, USA |
| **d** | Dead collected specimen |
| **L** | Length (mm) |
| **lv** | Live collected specimen |

| MNHNCL | Museo Nacional de Historia Natural, Chile |
| NDMP | Nazca Desventuradas Marine Park |
| NMW | National Museum of Wales, Cardiff |
| RAS | Richard A. Salisbury |
| ROV | Remotely operated underwater vehicle |
| SCBUCN | Sala de Colecciones Biológicas de la Universidad Católica del Norte, Chile |
| SDMNH | San Diego Museum of Natural History, San Diego, USA |
| W | Width (mm). |

## ACKNOWLEDGEMENTS

For their assistance at sea we would like to thank the Captain and crew of R/V Cabo de Hornos of the Chilean Navy, and the scientific personnel participating in the CIMAR 22 cruise. Special thanks also go to Erin Easton, Ariadna Mecho and Jorge Avilés for their help during collection, handling and curation of the specimens, and to Maria S. Romero for helping with the SEMs. We are grateful to Dr. Matthias Gorny from OCEANA who piloted the ROV that obtained the images of the habitat at the seamounts surveyed in this study. Marina Fuentes (MZUC), Oscar Gálvez (MNHNCL) and Guillermo Guzmán (Universidad Arturo Prat, Iquique, Chile) provided additional information and material on Chilean Mitridae. Our appreciation to John P. Wolff for checking grammar of this manuscript. We also acknowledge Dr. William "Bill" H. Clark, director of the Orma J. Smith Museum of Natural History for his contributions and help with this work. Bruce Marshall, Alexander Fedosov and an anonymous reviewer are also thanked for providing constructive insight that much improved the article.

### Funding

This work was supported by grants (CONA C22 16-09, FONDECYT 1181153, FONDEQUIP EQM 150109) and the Chilean Millennium Initiative, ESMOI. The funders had no role in study design, data collection and analysis, decision to publish, or preparation of the manuscript.

### Grant Disclosures

The following grant information was disclosed by the authors:
CONA: C22 16-09.
FONDECYT: 1181153.
FONDEQUIP EQM: 150109.
Chilean Millennium Initiative, ESMOI.

### Competing Interests

The authors declare that they have no competing interests.

## Author Contributions

- Javier Sellanes conceived and designed the experiments, performed the experiments, analyzed the data, authored or reviewed drafts of the paper, approved the final draft.
- Richard A. Salisbury conceived and designed the experiments, performed the experiments, analyzed the data, authored or reviewed drafts of the paper, approved the final draft.
- Jan M. Tapia performed the experiments, analyzed the data, prepared figures and/or tables, approved the final draft.
- Cynthia M. Asorey performed the experiments, analyzed the data, prepared figures and/or tables, authored or reviewed drafts of the paper, approved the final draft.

## Field Study Permissions

The following information was supplied relating to field study approvals (i.e., approving body and any reference numbers):

Sample collection was performed under permission Res. Ext N°41/2016 from SERNAPESCA (Chile) to Universidad Católica del Norte.

## Data Availability

The specimens for study are deposited in the following collections:

(1) Museo Nacional de Historia Natural (MNHNCL), Chile. Holotype MNHNCL 203730, Paratype 1 MNHNCL 203731, paratype 4 MNHNCL 203732.

(2) Academy of Natural Sciences of Drexel University (ANSP), Philadelphia, USA. Paratype 3 ANSP 476798.

(3) Orma J. Smith Museum of Natural History (CIDA), The College of Idaho, USA (CIDA). Paratype 2 CIDA 126,574.

(4) Sala de Colecciones Biológicas de la Universidad Católica del Norte (SCBUCN), Coquimbo, Chile.

Paratype 5 SCBUCN 7627, paratype 6 SCBUCN 6953, paratype 7 SCBUCN 7029, paratype 8 SCBUCN 7033, paratype 9 SCBUCN 7038, paratype 10 SCBUCN 6952a, paratype 11 SCBUCN 6952b, paratype 12 SCBUCN 7031, paratype 13 SCBUCN 7030, paratype 14 SCBUCN 6946a, paratype 15 SCBUCN 6946b, paratype 16 SCBUCN 6946c, paratype 17 SCBUCN 6946d, paratype 18 SCBUCN 6947a, paratype 19 SCBUCN 6947b, paratype 20 SCBUCN 6947c.

## New Species Registration

The following information was supplied regarding the registration of a newly described species:

Publication LSID: urn:lsid:zoobank.org:pub:787A4D2A-260C-49BC-B8B0-0665F2BF6108.

Atrimitra isolata sp. n. LSID: urn:lsid:zoobank.org:act:910607EF-EE88-47F9-A42D-21F2996E932A.

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
