# Peer review of "A new species of Atrimitra Dall, 1918 (Gastropoda: Mitridae) from seamounts of the recently created Nazca-Desventuradas Marine Park, Chile"

_PeerJ, doi:10.7717/peerj.8279_

## Round 0.1 · original submission · Major Revisions

The three reviewers suggested a major revision of the manuscript. I completely agree with their comments. Such data on organisms from remote habitats of the deep-sea are important contributions. The most substantial points that should be addressed from the reviewers comments are (1) a better account of the state of the art of the taxonomy (2) a more detailed examination and discussion of the relevance of morphological examined. Two of them also underlined that in a modern approach of species description for molluscs, it is a pity to not include molecular data when living animals have been sampled. I totally agree with this comment. Indeed basic molecular data would have provided a more robust positioning of the new species. Such an addition would substantially improve the manuscript. An additional comment is about the weak exploitation of the underwater images. Only a very basic description of the habitats and the distribution over the explored seamounts is provided. A better description of the habitats would be another substantial improvement of the manuscript.

·

Basic reporting

.

Experimental design

.

Validity of the findings

.

Additional comments

This manuscript is logically constructed, the science is sound and the illustrations are adequate.

I recommend it for publication after attention to the English, which will require quite a lot of work, and the following points:

38. NW in full.

50. The old “Turridae” is now recognised as representing numerous families. Very few if any of the species recorded here will actually be turrids. Either use the correct families or substitute Conoidea.

56. Should be Atrimitra orientalis (Gray, 1834)

57. Lower case v for von

112, 113. The diagnosis is very poor.

122. Last adult whorl preferable to (meaningless) body whorl.

Throughout. All coordinates should be in this format (e.g.) 25°46.65’S, rather than -25.7774, etc.

192. Suggest delete “that is” and “Dall, 1918” for clarity.

198. Should be … was listed as a synonym of Atrimitra idea by Cernohorsky (1976), but this has yet to be verified.

213. Should be … 180 and 280 metres on seamounts on rocky ground.

216. A. orientalis (Gray, 1834)

234. Delete word Mitridae

238-239. … cusps on the rachidian tooth.

248. Five

252. This limpet is a hipponicid, possibly a species of Malluvium.

·

Basic reporting

The records of the new species of the family Mitridae collected from the seamounts of Nazca-Desventuradas Marine Park are analyzed in the context of poor but highly endemic molluscan fauna of the studied region. Given that the areas of high local endemism host considerable fraction of yet undescribed marine biodiversity, investigation of such areas is important and the revised MS is a relevant and valuable contribution to our knowledge of habitats and fauna of one such area. The paper is written in decent English and is generally clear (although see specific comments to the text). The structure of the manuscript is straightforward, although can be further improved. The figures are sufficient for the topic – they clearly illustrate the new species, its distribution and similar species that authors find important to illustrate.

Experimental design

The discovered species of the family Mitridae is certainly new, and its taxonomic description is timely and well-tailored. Nevertheless, strictly speaking, the study does not follow best practices of the modern taxonomic research, and it is pity. The study is entirely dedicated to the description of one new species, and I find that 1) allocation of this species to genus (which is tentative, and authors admit it), would have been more convincing if addressed with molecular-phylogenetic techniques, and 2) a more comprehensive study of morphology would be desirable. Addressing the placement of a new species collected as recently as in 2016, with tissues obviously available for the DNA extraction, only by means of morphology (which is quite inconclusive) is clearly the main drawback of the present study. Unless the new species represents a separate lineage of the family (which would be a very interesting result by itself), the affinities to known taxa could have been demonstrated explicitly. Therefore, there would be no need to speculate about possible relatives that are as distantly related as Isara (subfamily Isarinae), and Atrimitra (subfamily Mitrinae). Furthermore, I find it strange that authors did not consider the deep water genus Calcimitra, with similar shell proportions and sculpture as a relevant placement for the new species. Besides, I find it strange that the authors, having life collected specimens in hands, refer to the poor resolution of the ROV photos as a reason why morphology of soft parts has not been studied!

Validity of the findings

The mentioned shortcomings do not undermine significantly the validity of finding - i.e. a new species is detected and described. The material available for the new species is clearly sufficient for description, and the conclusions are relevant and are backed up by the original data.

Additional comments

Attached is the submission PDF with my comments, which, I am convinced, will help authors to improve quality of the MS. In particular, I would encourage authors to:
1) State clearly how many specimens of the new species were available for study. How many of them were collected life, and how many were dead shells.
2) Provide a list of abbreviation of specimen repositories
3) Review description to remove references to methods of collection / imaging

Reviewer 3 ·

Basic reporting

The English seems correct throughout the manuscript. The structure of the article, the figures captions follow the format of the journal and are relevant to the content of the article.

The taxonomic background could be improved in the introduction, for instance how and why the species used for morphological comparisons were selected, while other where not (but see my further comments). In addition, there is a lack of consistency with the species names that are used throughout the text. For instance, some species names cited in the introduction are not valid (e.g. Mitra orientalis Griffith and Pidgeon, 1834 and Mitra semigranosa Von Martens, 1897, line 56 and 57), but become valid later in the text (e.g., Atrimitra orientalis, Line 167). Finally, the full names (including authors and year of descriptions) of the studied species do not appear at the first occurrence in the manuscript.

The morphology of the protoconch is used for the description of the new species and the SEM images are very interesting. However, nothing is said about the inferred larval mode of development. The SEM images clearly show a multispiral protoconch, which is indicative of a planktotrophic mode of larval development (even if the protoconch is eroded, I think this should clearly appear in the description), yet, the authors conclude that because the species was never found elsewhere in the Pacific, it must be endemic of these seamounts.

Experimental design

This is an original primary research that follows the Aims and Scope of the Journal, as it is Biological Sciences based on methodological soundness. All the field study permits seem to have been respected. Line 80: ‘Sample collection was performed under permission Res. Ext N°41/2016 from SERNAPESCA (Chile) to Universidad Católica del Norte.’

This paper fill the gap as by describing a new species, it increases our knowledge of deep-sea biodiversity. The investigation has been conducted rigorously but some information are lacking (notably about the taxonomic background and the hypothesis of endemism). Methods are described with sufficient details and information to replicate.

Validity of the findings

I believe that the description of the species is sound and necessary but that some statement about its geographical distribution should be exposed with more caution. The authors selected a lot of paratypes that are deposited in many institutions. These specimens have been fixed in Ethanol which should provide good data for further studies. Some of the specimens are lacking ID numbers. These numbers should be obtained before the final acceptance of the manuscript.

Additional comments

The manuscript describe a new species of Mitridae, Atrimitra isolata sp. n., collected from seamounts located near Desventuradas Islands, off Chile. The description is based on one holotype and twenty paratypes collected by trawling on three seamounts (SF5, SF6 and SF9). The authors tentatively provide some insights into the habitat of this new species based on underwater imagery taken with a ROV on these same seamounts plus an additional one (SF2), which could not be trawled because of the roughness of the terrain. The species description is based on the analyses of the general shell morphology, and on the examination of the protoconch and the radula using scanning electron microscope (SEM).
Shell morphological characters are compared to three congeneric shallow water species from California (holotype of Atrimitra idea (Melvill, 1893)), Chile (Atrimitra semigranosa (von Martens, 1897)) and Peru (Atrimitra orientalis (Griffith and Pidgeon, 1834)) and a fourth species (holotype of Strigatella (Atrimitra) coronadoensis Baker and Spicer, 1930) from Mexico, for which the current taxonomic status is unknown. Although an effort was made to gather this material for morphological comparisons, nothing is said about why other congeneric species from the same geographical area were not examined. For instance Atrimitra effusa (Broderip, 1836) from Central America and Galapagos islands, and Atrimitra catalinae (Dall, 1919) from California.
Given the recent publication of Fedosov et al., 2018 on the molecular systematic and morphology of the Mitridae, which is based on four genes but has no material representing the Atrimitra genus, it is a pity that this fresh material was not used to publish some DNA sequences. However, the morphological comparisons seem robust enough to describe these specimens as belonging to a new species. For that matter, I think that the manuscript has potential to be of interest to readers of the journal. However, I have identified some major problems and advise the authors to revise their manuscript.
The first major problem is the taxonomic background that I think could be improved in the introduction and throughout the manuscript. The second major problem concerns the mode of larval development that is overlooked although the protoconch is examined. Moreover, conclusions are made about the endemism of this new species without any consideration about its potential larval mode of development and the global the lack of knowledge on the deep-sea biodiversity. Please see my comments below for further details.

Detailed comments.
ABSTRACT:
The abstract is quite short, could the authors be more specific here about which characters (at least the most obvious) make them say that the new species is ‘morphologically related to counterparts (..) from the west coast of North, Central and South America’ but ‘has no affinities with species of the family found around Easter Island, ….’?
INTRODUCTION:
Line 49-50: please homogenize the sentence by using either Latin or vernacular names: i.e. Polyplacophora vs. gastropods, bivalves and cephalopods.
Line 56: The species name Mitra orientalis Gray, 1834 is no longer accepted, please use the valid species name: Atrimitra orientalis (Gray, 1834), or specify in the text that the name has changed.
Line 57: The species name Mitra semigranosa Martens, 1897 is no longer accepted, please use the valid species name: Atrimitra semigranosa (Martens, 1897) or specify in the text that the name has changed.
Lines 56-57: The full names (including authors and year of descriptions) of the studied species do not appear at the first occurrence in the manuscript. Some species name used in the introduction are taxonomically not valid (e.g. Mitra orientalis Griffith and Pidgeon, 1834 and Mitra semigranosa Von Martens, 1897) lines 56 and 57). Even if the correct names are used later in the text, the current taxonomic status of all the species analyzed should be more clearly introduced, in the Introduction section or, as suggested in the Guidelines of the Journal, in a Table.

MATERIALS AND METHODS:
Line 79 & Line 81-82: The definition of SEM (scanning electron microscope) should be given at the first occurrence (line 79).

RESULTS:
Line 115-120: Although the protoconch is highly eroded (i.e., top of the protoconch I is missing and protoconch II lack of sculpture…), it is obviously a multispiral protoconch, which is a reflexion of planktotrophic larval development. I would suggest the authors to highlight more clearly this point about the larval development. This could be interesting in the following discussion about endemism and the potential distribution of that species.

Line 142, 143 and 146: please provide an ID numbers (CIDA, ANSP and SCBUCN) before acceptance of the ms.
Line 192: please provide the full species name (i.e. with author and date) of Atrimitra idea (Melvill, 1893) at its first occurrence in the ms, which is Line 163.
Line 164 and caption in Figure 4: please provide the full species name (i.e. with author and date) of Strigatella coronadoensis. This species name does not appear as valid. Please provide more information about the currently accepted name for this species. .. Some clarification appear Lines 197-199 but perhaps a bit late. I suggest clarifying this in the introduction. (i.e. the choice of the species used for comparison and an overview of their current status).
Line 216: Same comment for Atrimitra semigranosa (Martens, 1897), which first occurrence appears line 165.
Line 168: Same comment for Atrimitra orientalis (Gray, 1834).
Line 105: I suggest removing the subfamily rank as it is currently unassigned to the genus Atrimitra. Please check out WoRMS : http://www.marinespecies.org/aphia.php?p=taxdetails&id=411799
And http://www.marinespecies.org/aphia.php?p=taxdetails&id=1027672

DISCUSSION:
Line 230-232: ‘The new species is isolated from the mainland and seems to be endemic to the Nazca Plate, where it lives in deep water and associated with seamounts’. Given the paucity of the deep-sea biodiversity exploration and the larval development of the described species, I would suggest the authors to be more cautious when stating that this new species might be endemic.
Line 238-250: It is not clear what drawings the authors are talking about. It is mentioned Line 240: ‘only drawings of the radula have been published (Fedosov et al., 2018).’ However, no drawing was produced in Fedosov et al., 2018. Are the authors referring to older description such as Melvill, 1893 or Cernohorsky (1970)? In that case, the in-text citing should be more precise.

Line 248-249: ‘However, there are not enough SEM images of radulae of this type (see Fedosov et al., 2018)’: again, can the authors be more specific about what part of Fedosov et al. 2018’s paper they are referring to? This paper did not use Atrimitra material and consequently did not provide any diagnosis for that genus.

Line 264: ‘The new species has no affinities with them”: It is not clear what the authors are meaning here. What kind of affinities are the authors referring to?

Line 265: ‘available evidence suggests that it is found only on these seamounts’: What evidence is the authors referring to? Also, as mentioned earlier (Line 230-232 comment), given the paucity of the deep-sea exploration, and the larval development of the described species, which clearly is planktotrophic, the authors should be less assertive when stating that this species is endemic.
Conclusion about

REFERENCES:
Line 246, 254, 257: the authors used “pers. obs.”. As recommended in the PeerJ guidelines to authors Reference as 'pers. comm.', must include the name and year.
General remark: I did not check them all, but some available DOIs are missing from the reference list. For instance:
Lines 318-320 the doi is: https://doi.org/10.1093/zoolinnean/zlx073
Lines 331-332 the doi is : 10.3897/zookeys.219.3994

FIGURES:
Figure 2: CIDA XXXX please provide an ID number

Annotated reviews are not available for download in order to protect the identity of reviewers who chose to remain anonymous.

---

## Round 0.2 · Minor Revisions

The manuscript is now sufficiently improved to be published. However, please take into account A. Fedossov's comments by (i) clarifying what you mean by "DNA integrity". (ii) providing more details about the method of preservation of specimens and the molecular test about the quality of the DNA.

As noted by the second reviewer, please try to improve the English notably in the introduction and discussions sections.

·

Basic reporting

I find that the English improved sufficiently and the paper can be accepted in its current state

Experimental design

Experimental design did not change since first submission

Validity of the findings

The findings remain valid, however I am convinced that authors should make efforts to obtain sequencable material of Atrimitra isolata in future.

Additional comments

I am satisfied with the revisions that authors have made, and recommend the paper to be accepted for publication. Nevertheless, I would like to still make some comments that may be useful for authors in their future studies. 1. Breaking shell is not necessary to find out whether the specimen is life taken or dead. Drilling a hole ~1.5 whorls above aperture would be enough, and the dentistry drills and bores perform really well. 2. I did not understand what did author mean under DNA intergrity? If they assert that the extraction of DNA was 'effective' - how then the integrity test was 'negative'? What do author mean by that? This response is very vague, and I would like to have details - have the authors checked the total DNA quality by gel-electrophoresis, and what results were like? What primers were used in the attempted PCRs? 3. Do authors know, how the bulk specimens were preserved? I.e. whether they were fixed with Formaldehyde or ethanol? If by the formaldehyde, announced future attempts with the use of forensic techniques won't likely be efficient either.

Reviewer 3 ·

Basic reporting

The English could be improved in the Introduction and Discussion sections.

Experimental design

no comment

Validity of the findings

no comment

Additional comments

The manuscript has been largely improved since its first submission and it seems that many of the reviewers’ comments have been integrated to the present version.

I only have minor remarks concerning the English writing and the structure of some parts, which could be improved notably in the Introduction and the Discussion sections. Please see my detailed comments bellow.

Therefore, for the comfort of the reader and for the clarity of the text, my recommendations is that the manuscript should be thoroughly red by an English native speaker and reviewed accordingly before final acceptance and publication in PeerJ.

Detailed Comments:

INTRODUCTION
Line 45: Add « the » before « estimated endemism »
Line 47-49: This is an awkward sentence. I suggest:
“Conversely, information on the invertebrates of the area is sparse. Most of the existing references are associated with research expeditions carried out between 1973 and 1987 by the former Soviet Union and limited to the area beyond Chilean jurisdiction east of ~83°W.”

Line 51: “in general” is repeated twice.

Line 52: remove « s » to « report »
Line 53: replace « a polyplacophoran » by « one polyplacophora species »
Lines 52-55: It is not very clear. Are the 42 molluscan species listed here endemics of the SYGN ridges? Or do they represent the total number of mollusks reported in the area?
If they are endemics, an easy way to formulate would be: “For mollusks, these authors reported that 42 species are endemic of the 22 seamounts along the Salas y Gomez and Nazca ridges explored”
Lines 53-54: As suggested in my previous review, taxa name here could be homogenized as follow: one polyplacophora species, 27 species of gastropoda, seven species of bivalvia, and seven species of cephalopoda.
Line 55: “the latter most probably pelagic”: I do not think that “pelagic” is the right word. It really depends of the species but I believe these cephalopoda to be sedentary, suprabenthic or epibenthic. (i.e. swimming bottom-dependent animals living in the water layer just above the sea floor).
Line 55: “In this,” is cryptic… and “as well as in subsequent studies of mollusks of the area” is a bit awkward. I would suggest: “In Parin et al. (1997) and subsequent malacological studies in the area, no representatives of the family Mitridae have ever been mentioned.”
Line 57-59: awkward sentence. I would suggest: “However, in the westernmost side of the Salas y Gómez ridge, at Rapa Nui (Easter Island), Osorio (2018) mentioned the occurrence of the three following Mitridae species: Strigatella flavocingulata (Lamy, 1938), Imbricariopsis punctata (Swainson, 1821) and Neocancilla takiisaoi (Kuroda, 1959).”

MATERIALS AND METHODS:
Line 100-103: As mentioned in my first review, the definition of SEM (scanning electron microscope) should appear at the first occurrence (line 100).
Line 105: Replace “cleaning” by “cleaned”

RESULTS:
Line 158: “L: 20.4 mm, W: 7.3., AL: 10.2”: Either keep or remove the “mm” for all L, W and AL.
Line 158: “10.2; Seamount”: Replace the semi-colon by a colon.
Lines 159, 193, 197, 198, etc.: “Lat. -25.7774°, Long. -83.163°”: If coordinates are given in decimal then the degree, (°), should be removed.
Line 160: I do not understand what this mean: “specimen 3 of 6, C22 SSF9 A,”
Line 186: As mentioned in my previous review, what is the current valid name of “Strigatella coronadoensis, holotype SDMNH 44409-667”?
Line 187-189: Remove the spaces after “(Fig. 4A–C)” “(Fig. 4A–C)” and “(Fig. 4G–I)”.

DISCUSSION:
Line 251: Atrimitra isolata sp. n. is one of the only few Mitridae reported from Chilean waters
Line 252: remove the “it” -> “so far has been”. Replace “at” by “on”: “only on the Nazca”.
Line 253: I am not sure but I think that here “with” should be replaced by “to”: “associated to seamounts”. A more easy way to say that would be: “where it lives on deep-water seamounts”.
Line 255: the “Although” here make your sentence unfinished.

Line 291: replace “live” by “living”
Line 299: Replace “ecologic aspects” by “ecological aspects”
CONCLUSION:
Line 304: Atrimitra isolata should be in italic
Line 309: Atrimitra and Isara should be in italic

GENERAL REMARKS:
Lines 278, 294, 297: the authors used “pers. obs.”. However, as recommended in the PeerJ guidelines to authors Reference as 'pers. comm.', must include the name and year.

---

## Round 0.3 · Minor Revisions

The article is very close to Acceptance. However, the authors did not really explain the problem encountered for obtaining molecular data. Considering the state of knowledge, the description of a new species in these taxa should when possible include such data. Therefore you should explain in the manuscript why their obtention failed - at present you have simply avoided the question.

---

## Round 0.4 · accepted · Accept

The present version correctly explain why molecular data are not available. I am thus pleased to accept the paper.